


# The effects of storms and a transient sandy veneer on the interannual planform evolution a low-relief coastal cliff and wave-cut platform at Sargent Beach, Texas, USA

Rose V. Palermo[1,2], Anastasia Piliouras[1,3], Travis E. Swanson[1,4], Andrew D. Ashton[5], David Mohrig[1]

[1] Department of Geological Sciences, Jackson School of Geosciences, The University of Texas at Austin, Austin, 78712, USA
[2]Joint Program in Oceanography/Applied Ocean Science and Engineering, Massachusetts Institute of Technology/Woods Hole Oceanographic Institution, Woods Hole, 02543, USA
[3]Earth and Environmental Sciences Division, Los Alamos National Laboratory, Los Alamos, 87545, USA
[4]Department of Geology and Geography, Georgia Southern University, P.O. Box 8149, Statesboro, 30460, USA
[5] Department of Geology and Geophysics, Woods Hole Oceanographic Institution, Woods Hole, 02543, USA

*Correspondence to*: Rose V. Palermo (rpalermo@mit.edu)

**Abstract.** Coastal cliff erosion is alongshore-variable and episodic, with retreat rates that depend upon sediment as either tools of abrasion or protective cover. However, the feedbacks between coastal cliff planform morphology, retreat rate, and

sediment cover are poorly quantified. This study investigates Sargent Beach, Texas, USA at the annual to interannual scale to explore (1) the relationship between temporal and spatial variability in both cliff retreat rate and roughness and (2) the response of retreat rate and roughness to changes in sediment cover of the underlying mud substrate and the impact of major storms, using the low-lying mudstone cliff as a rapidly evolving model of a larger cliff system. A storm event in 2009 increased the planform roughness and sinuosity of the coastal cliff at Sargent Beach, TX. Following the storm, satellite

image-derived shorelines with annual resolution show a decrease in average alongshore erosion rates from 4 to 12 m yr-1, coincident with a decrease in shoreline roughness and sinuosity (smoothing). A storm event in 2017 again increased the planform roughness and sinuosity of the cliff. The occurrence of storms and the presence of sediment to laterally erode the cliff influence the planform morphology and subsequent retreat. Over shorter timescales, monthly retreat of the sea cliff occurred only when the platform was sparsely covered with sediment cover on the wave cut platform, indicating that the

tools and cover effects can significantly affect short-term erosion rates. The timescale to return to a smooth shoreline with a long-term steady-state erosion rate following a storm or roughening event is approximately five years, with the long-term rate suggesting a minimum of ~38 years until Sargent Beach breaches, compromising the Gulf Intracoastal Waterway (GIWW) under current conditions and assuming no future storms or intervention. The observed retreat rate varies, both spatially and temporally, with cliff face morphology, demonstrating the importance of multi-scale measurements and

analysis for interpretation of coastal processes and patterns of cliff retreat.



## 1 Introduction

Coastal cohesive clay cliffs may recede at rates of meters per year or more (Sunamura, 2015) depending on the intensity of waves, sea level rise, and the tools and cover effects of sediment abrasion (Sunamura 1992; Sunamura, 2015; Limber and Murray, 2011; Young et al., 2014). Soft sediment cliff erosion is variable and episodic in the alongshore direction
(Sunamura, 2015), and internal sediment dynamics play an important role in the alongshore cliff morphology (Limber and Murray, 2011). However, the feedbacks between storms, sediment cover, and planform morphology of coastal cliffs remains poorly quantified (Limber and Murray, 2011, Limber et al., 2014, Sunamura, 2015). In the coming years, these coastal erosion processes along with climate-change-driven increases in hazards pose an increasing threat to coastal communities and infrastructure (Oppenheimer et al., 2015).


Sargent Beach, Texas, USA, (Fig. 1) is a consolidated Holocene mud beach composed of floodplain sediments that outcrop as a low-relief sea cliff and Type-A (Sunamura, 1992), gently sloping wave cut platform, ephemerally covered by sand and shell hash. Sargent Beach is found in a 17 km stretch of coast eroding at an average of 15 m yr$^{-1}$ over the last three decades, the largest concentrated extreme of shoreline erosion globally (Luijendijk et al., 2018). This small and dynamic system can
be examined as a model for erosion of larger coastal cliff systems, allowing us to understand and explore the feedbacks between planform morphology and the evolution of cliffs over time scales of months to years. Similar cohesive coastal cliffs exist globally, including in the Caribbean coast of Colombia (Paniagua-Arroyave et al., 2018), Lake Michigan, USA (Brown et al., 2005), and London, UK (Hutchinson, 1973).

Although shoreline change at Sargent has been historically analyzed using measurements spaced 50 meters or more apart and averaged over decades, these observed shorelines do not capture change at the scale of the embayments in the cliff face or changes in response to tropical storm seasons (Sealy and Ahr, 1975; Morton, 1977; Stauble, et al., 1991; Paine et al., 2011, 2014). We study this landscape at length scales of tens of meters and timescales of months to years to describe and understand the mechanisms of erosion that drive the high retreat rates at Sargent Beach. Our measurements show that storm
events increase the roughness and sinuosity of the shoreline, which drives high rates of erosion for years afterward. Monthly measurements allow us to evaluate the relationship between sediment cover and cliff face retreat rate and morphology. We use the cliff face morphology and retreat rates to evaluate the temporal and spatial relationships between roughness and sinuosity with storm events, sediment cover, and each other. These spatial relationships give us insight into the processes of erosion driving the localized high retreat rates at Sargent Beach and other larger coastal cliff systems. Further, investigating
the retreat of the low-relief sea cliffs at Sargent is important for the local community, as it is the barrier between the Gulf of Mexico and the Gulf Intracoastal Waterway (GIWW). Because of its thin and narrowing nature, its sediment-starved character, a consistently high erosion rate, and the recurrence of hurricanes and tropical storms, Sargent Beach is at risk of breaching on a foreseeable timescale. This breaching would have major economic and environmental repercussions.





## 2 Background

### 2.1 Sargent Beach Setting

Sargent Beach, Texas, USA sits on a narrow, 150 m strip of barrier coast that separates the GIWW (Fig. 1). Alongshore sediment transport at Sargent Beach is directed from the northeast to the southwest and the mean tidal range is 20 cm (NOAA tide station 8772985). Ephemeral sand is underlain by Holocene consolidated mud that commonly outcrops in the surf zone (Fig. 2A) (Paine et al., 2014). Inspection of cliff retreat maps of the Texas Gulf Coast reported in Paine et al. (2014) shows that the highest shoreline retreat rates occur at sections of the coast with exposed mud in the surf zone. Since 1856, the local shoreline has undergone 740 m of landward retreat, approximately a 5 m yr$^{-1}$ long-term retreat rate, putting the GIWW, a major inland barge-transportation route, at a high risk of breaching.

The Holocene mud substrate is composed of subhorizontal beds, centimetres to decimetres in thickness, with varying densities of preserved plant roots. This muddy substrate consists of floodplain and marsh deposits from the Caney Creek overbank system which was the larger Colorado River prior to its most recent avulsion and establishment of the modern river pathway (McGowen et al., 1975; McGowen and Macon, 1976). Compressive strengths for the Holocene mud substrate were estimated in the field using a Forestry Suppliers Pocket Penetrometer and range from 412 kPa for dry mud to 206 kPa for moist mud and very weak for submerged, fully saturated mud. The mudstone substrate is sculpted into a wave-cut platform that often terminates at a low-relief sea cliff when the difference between local elevation of the Holocene mud substrate and sea level is sufficiently large (Fig. 2B) (Bradley, 1958; Stauble, et al., 1991).

"Nourishment" projects (placement of fill on the beach) have been implemented along Sargent Beach to mitigate extreme coastal erosion caused by large storms and an interruption of littoral drift by the protruding Brazos River delta that has been hypothesized to starve this section of coast of sand (Seelig and Sorensen, 1973; Morton, et al., 2004). In 1988, a combination of mud and sand dredged from the Gulf Intracoastal Waterway was emplaced on Sargent Beach in an effort to counteract cliff retreat. Most of this sediment was transported away from the nourishment site within one year (Morton and Paine, 1990). Another beach nourishment project was commissioned in 2013 which added 66,723 m$^3$ of sand onto Sargent Beach (Bush, 2015). Although average shoreline retreat rates are decreasing on the Texas Gulf Coast (Paine et al., 2014), despite these nourishment projects, at Sargent Beach shoreline retreat remains high.

### 2.2 Storm history

Over the past decade, Sargent Beach has experienced several intense storms, including Hurricane Ida in 2009, Tropical Storm Bill in 2015, and Hurricane Harvey in 2017. Hurricane Ida travelled north-northwest across the Gulf of Mexico towards the mouth of the Mississippi River in November 2009 as a tropical storm and later a Category 2 hurricane, before turning east and making landfall in Alabama (Avila and Cangialosi, 2010). Tropical Storm Bill made landfall near





Matagorda Island on June 16, 2015 (Berg, 2015). Hurricane Harvey first hit the Texas Gulf Coast ~150 km southwest of Sargent Beach on August 26, 2017 at Category 4, and returned to the Gulf ~65 km southwest of Sargent Beach on August 28 as a tropical storm passing over Matagorda Bay – a site less than 30 miles from Sargent Beach (Blake and Zelinsky, 2017). These storms each produced large storm surge and waves that knocked out buoys and eroded Sargent Beach over the study
period.

## 2.3 Coastal erosion processes and sea cliff evolution

Substrate erosion at Sargent Beach occurs through a number of processes that have been observed at other coastal cliff settings, and is dominated by abrasion from water-entrained siliciclastic sand and shell fragments, as well as repeated wetting and drying of the foreshore substratum that causes polygonal fracturing, promoting quarrying of small mud blocks
that in turn rapidly disaggregate into their constituent grains (Fig. 3a) (Anderson, 1986; Trenhaile, 1987; Hancock et al., 1998; Stephenson and Kirk, 2000; Stock et al., 2005). Abrasion of mud by shell hash and sediment occurs through four distinct styles of focused erosion. First, focused impact of grains on the sea-cliff base and energy dissipation from wave impact leads to undercutting at the toe of the cliff and subsequent gravity-induced failure of the overhanging cliff face (Brooks et al., 2012; Collins and Sitar, 2007; Kline et al., 2014; Quinn et al., 2010, Adams et al., 2005). Cliff retreat by such
failures maintains a vertical cliff face through time (Gardner, 1983) (Fig. 3b). Second, the swash and backwash motion of water-entrained grains cuts grooves or runnels into the platform that are oriented roughly perpendicular to the shoreline and parallel to the direction of swash and backwash (Fig. 3c). These runnels develop because of the feedback between topography and erosion rate, brought about by a focusing of the concentration of abrading particles within the linear troughs (Allen, 1987; Fagherazzi and Mariotti, 2012; Flood, 1983; Williams et al., 2017). Third, outsized pebble clasts grind
potholes into the mud substrate (Fig. 3d) (Pelletier et al., 2015). Finally, a substratum consisting of subhorizontal beds with different erodibilities leads to the production of discrete, seaward-facing steps, centimetres to decimetres in relief (Fig. 3e). These steps, potholes, and runnels are cut into a landward-migrating and gently seaward dipping wave-cut platform at Sargent Beach.

On rocky coasts (including those composed of consolidated mud), the amount of sediment covering the foreshore and shoreface plays a key role in determining both the magnitude and pattern of substratum erosion (Sunamera, 1976; Robinson, 1977; Walkden and Hall, 2005; Limber and Murray, 2011; Young et al., 2014). Loose, mobile sediment can either facilitate this erosion by acting as abrasional tools or inhibit erosion by mantling and protecting the vulnerable mud substrate (Sunamera, 1976; Robinson, 1977; Sklar and Dietrich, 2001, 2004; Walkden and Hall, 2005; Limber and Murray, 2011). The
tools and cover effects have been widely studied in the context of bedrock river incision, but largely overlooked for bedrock or mud substrate beach erosion under the influence of wave oscillation (Bramante et al., 2020).

Sediment availability at Sargent Beach is insufficient to completely cover the foreshore at all times. This sediment-limited environment allows sediment particles act as tools of abrasion. The scarcity of loose sediment at Sargent Beach is linked

both the local mudstone lithology and its position 19 km down-shore from the trailing edge of the modern Brazos River delta, which disrupts the littoral cell and captures the alongshore transported sand as it grows seaward (Seelig and Sorensen, 1973; Morton, et al., 2004). Meanwhile, constant wave action and little sediment supply result in a persistent erosion of Sargent Beach (Morton and Pieper, 1975; Morton, 1979).

Headland coasts tend to become less complex and straighter through time as headlands erode and bays fill in with eroded sediment (Trenhaile, 1987; Trenhaile, 2002; Valvo et al., 2006; Limber et al., 2014). Rocky sea cliffs have been shown to decrease in spatial variability of retreat rate through time, though this has not been quantitatively replated to changes in shoreline roughness (Sunamura, 2015). Additionally, soft cliff retreat rate has been directly linked to wave height (Brown et al., 2005). However, alongshore coupled models show that rocky coastlines can reach an equilibrium configuration where

headlands and embayments remain stable over millennial timescales (Limber and Murray, 2011). Additionally, cliff erosion is episodic both temporally and spatially (Sunamura, 2015), and is at least partially controlled by sea level rise (Ashton et al., 2011).

### 2.4 Outline

Here, we conducted a series of investigations of a rapidly evolving cohesive coast to study the dynamics across annual,

monthly, and storm event scales to better understand cliff erosion process in the context of longer-term evolution. To evaluate the feedbacks between cliff face morphology and retreat rate, we use annual aerial images to digitize the cliff face and quantify the morphology and retreat. Monthly field surveys of sediment cover and cliff retreat give insights into the controls on erosion and morphology that sediment has in this system. Finally, we use lidar of the cliff before and after Hurricane Harvey to study the effect a single major storm can have on cliff morphology.

## 3 Methods

Two field sites were chosen at Sargent Beach to compare erosion mechanisms, rates, and morphologies of wave-cut platforms with sea cliffs (Fig. 1). Sargent Beach's sea cliff is located at Site 1 (Fig. 1). Site 2 is on the wave-cut platform down-shore from Site 1 (Fig. 1, 2).

### 3.1 Remote sensing

For the remote sensing analysis, we used ~annual aerial images with 0.5 m resolution over the years 2009, 2010, 2011, 2012, 2014, 2015, and 2017 (Fig. 4, Table S1). We manually traced the most landward position of the cliff face, identified visually



by demarking either the contact between sediment armouring the wave-cut platform below and the cliff, the contact between the cliff and the water, or the stark relief.

For the sea cliff, we calculated local retreat rates at one meter alongshore intervals as the change in cliff position perpendicular to the linear regression of all mapped cliff faces. The detection limit based on pixel size (0.5 m) and georeferencing error (calculated for each image pair) both contribute to the uncertainty of calculated cliff retreat rates. This uncertainty is computed using the apparent displacement of single stationary structures in the images (i.e., the georeferencing error) and the pixel size.


In the same one meter increments along the exported cliff faces, we calculated local roughness (m) as the absolute value of the localized difference between the linear regression and the cliff face position (Fig. S3-S4). Similarly, pixel size comprises the roughness uncertainty. We also calculate the sinuosity of the cliff face for each image, which is linearly correlated with roughness.


To evaluate three-dimensional changes in sea cliff morphology due to Hurricane Harvey, we analyze digital elevation models (DEMs) derived from airborne lidar, collected by the United States Army Corps of Engineers in 2016 and by the Bureau of Economic Geology at the University of Texas in 2017, before and after Hurricane Harvey, respectively (USACE 2016; Bureau of Economic Geology Preliminary Post-Harvey Survey Map). We compare transects of both the sea cliff and

the wave-cut platform and define the mean elevation of each feature as the characteristic elevation of that morphology for Sargent Beach. We also compare transects of Site 1 from before and after Hurricane Harvey to observe topographic change of the sea cliff. Vertical uncertainty in these lidar datasets is 0.2 m (USACE 2016; Bureau of Economic Geology Preliminary Post-Harvey Survey Map).

### 3.2 Field study

To measure short-term changes and to ground truth the remotely sensed data at both field sites, we conducted frequent elevation and local erosion surveys every 6 to 8 weeks throughout 2015 (Fig. 1). We measured 10 to 15 elevation survey transects perpendicular to the shoreline, with approximately 15 m spacing, beginning at the edge of the berm and extending approximately 30 m into the swash zone using a total station. Total station error is millimeter scale, negligible relative to other sources of uncertainty. Each survey point was identified as mud substrate, mobile overlying sand, or a transition

between the two. At Site 2, we used the total interpolated area of the study site and the areas of exposed mudstone substrate and sediment cover to calculate sediment cover as a percentage of the entire surface.

To determine local cliff face erosion rates, we placed erosion pins (15.2 cm screws) flush against the cliff face, approximately 0.5 to 1m up from the base of the cliff, on several locations perpendicular to, parallel to, and oblique to the

Earth **Surface**
**Dynamics**
Discussions

best-fit shoreline trend to capture the spatial variability of erosion. To measure the distance the cliff face retreated locally

between surveys, we measured the length of the erosion pin exposed for each subsequent survey (Fig. S1, Table S2).

To derive lateral retreat of the wave-cut platform, we measured a vertical lowering of the platform using an Army Corps of

Engineers Survey Mark located in the swash zone of Site 2 (See Fig. 1 for location of Site 2; Fig. S2). In 1990, this survey

mark was emplaced flush with the horizontal surface that is now the wave-cut platform. Vertical lowering and platform slope

measured in the field surveys were used to calculate lateral retreat.

## 4 Results

### 4.1 Sea cliff: retreat rates, roughness, and sinuosity

Local sea cliff erosion was spatially and temporally variable, with the promontories experiencing higher erosion rates than

the embayments (Fig. 4, 5). Both the mean and the standard deviation of retreat rate decreased through time; and this

decrease is greater than the measurement uncertainty (Fig. 5, 6a).

The retreat rate decreases steadily with time, starting high after Hurricane Ida in 2009 (Fig. 6a). Roughness and sinuosity

both increase following Hurricane Ida and Hurricane Harvey (Fig. 6b-d). In the years following these storms, subsequent

trends show that roughness decreased with time (Fig. 6b). Later years contained fewer sea cliff protrusions, lower roughness

values, and lower local erosion rates. Hurricane Harvey (2017) increased the roughness and sinuosity of the cliff face, but

not enough to significantly increase the retreat rate.

### 4.1.1 Modelling sea cliff steady-state retreat rate

We perform a non-linear least squares fit of the decay of retreat rate ($r$) through time using an exponential model, $r(t) =$

$ae^{-bt} + c$. Model fitting to the pre-Harvey roughness timeseries (Fig. 6a) yields fit values where parameter $a$ is 8.244 m yr$^{-1}$, $b$ is 0.969 yr$^{-1}$, and $c$ is 4.118 m yr$^{-1}$ ($c$ is the steady-state retreat value). This model fits our data with an R-square value of

0.9678. Using the fitted empirical model, we calculate a recovery timescale $t_r$ (i.e., the time to return to steady-state

conditions, or attain 99% of the fitted steady-state retreat rate $c$) by setting $r(t_r) = 0.99c = ae^{-bt_r} + c$ and solving for $t_r$.

### 4.2 Comparison of cliff retreat rates and sediment cover on the platform

There is little to no retreat for sediment cover >90% (Fig. 8). All erosion pins were lost between April and May, which we

interpret to represent an amount of erosion greater than or equal to the length of the pins (15.2 cm). For this time interval, the

plotted value represents the minimum retreat that occurred in this period, which was 0.054 m month$^{-1}$. The erosion pins were

buried via sand deposition that covered the cliff face between May and July, indicating no measurable erosion. The measured





retreat rate from July to September was 0.005 m month$^{-1}$. The erosion pins were lost again between September and
November, indicating a minimum retreat rate of 0.05 m month$^{-1}$.

## 4.3 Effects of Hurricane Harvey on the sea cliff

There is approximately a 0.5 m to 1.5 m $\pm$ 0.2 m difference between the elevation of the top of the sea cliff and the elevation
of the wave-cut platform, as determined from 2016 lidar data (Fig. 9a). During Hurricane Harvey, the storm surge at Sargent
Beach was recorded between 1.2 m to 2.1 m, which would have inundated the cliff (Blake and Zelinsky, 2017). After
Hurricane Harvey, the sea cliff shows development of a second step in its topography with 0.8 m $\pm$ 0.2 m in relief (Fig. 9b).
This change in topography is also seen in the 0.3 m contours of topography derived from the 2017 lidar (Fig. 9b). Two sets
of tightened contours are present in the cross-shore direction in the 2017 data, representing the differential vertical erosion
that developed the second step (Fig. 9b).

We differenced the lidar-derived DEMs from 2017 and 2016 to find the areas of most topographic change at Site 1 (Fig. 9c).
Evidence of overwash and washover deposits in the aerial imagery correspond to areas of accretion in the differenced lidar
image. Maximum erosion occurred in the embayments and hollows of the cliff. Here, sediments easily accumulate and are
used as tools of abrasion when entrained. Mud substrate relief was diminished after Harvey due to vertical lowering of the
sea cliff itself. Additionally, the cliff face erosion in the alongshore direction, or lateral erosion, notably increases the width
of embayments.

The mud substrate survey points allowed us to measure a shore-perpendicular wave-cut platform slope of 1.15°. Between
February 8$^{th}$, 2015 and November 15$^{th}$, 2015, we measured a vertical lowering of the platform of 15 cm and constant
platform slope (1.15°) at the USACE survey mark. During the 2015 surveys, we measured a minimum of 0.23 m of lateral
retreat of the sea cliff using erosion pins, while lateral retreat of the platform was estimated as 5 m using the USACE survey
mark and platform slope. This is an order of magnitude difference between cliff retreat and platform retreat over the same
survey period and less than two kilometers in alongshore distance.

## 5 Discussion

Changes in sediment cover are exogenic to coastal erosion at Sargent Beach, and instead are driven by changes in sediment
supply from storms, offshore, or up-coast. However, the amount of sediment in the system influences the morphology of the
cliff face. Here we show that cliff retreat occurs when sediment cover is <90% (Fig. 8). Additionally, our observations of
cliff morphology can be linked to erosion by sediment abrasion (Fig. 11). If sediment cover is 90% or greater, no cliff
erosion occurs and there is no change in morphology (Fig. 11b). When there is less than 90% sediment cover, but enough to
act as tools of abrasion, erosion is focused on the sides and back of the embayments, increasing the sinuosity and roughness





of the cliff face (Fig. 11c). When there is not enough sediment cover to act as tools, erosion by waves preferentially erodes the headlands, reducing the sinuosity and roughness of the cliff face (Fig. 11d). Because there is no feedback between erosion rate and sediment cover, as mudstone eroded from the cliff quickly disaggregates and leaves the system as washload, storm occurrence controls erosion at Sargent via controls on both wave activity and sediment supply.

Hurricane Harvey was the most recent major storm to impact the area, making landfall on the Texas Gulf Coast in 2017. The sea cliff at Sargent Beach lost much of its form due to erosion during Hurricane Harvey. High storm surge during Harvey resulted in waves that overtopped the sea cliff and eroded vertically down rather than landward, as evidenced by the vertical step in the former sea cliff (Fig. 9). Erosion due to Hurricane Harvey increased both the sinuosity and roughness of the cliff face. The observed sinuosity increase is attributed to erosion on the sides of the headland (Fig. 9), where sediment abrasion

may be most efficient (Fig. 11c). Sediment cover is often highest on the cliff face at Sargent Beach in the embayments and decreases to little or none in front of the headlands. The spatial patterns and variability in sediment transport alongshore may play a critical role in determining where the peak erosive efficiency may be for sediment as tools of abrasion in larger cliff systems. This may have a larger control on sinuosity and roughness of cliff faces than previously expected, given the importance of sediment cover in this system.


In June of 2015, Tropical Storm Bill made landfall on Sargent Beach, the only major storm during this study's field campaign. Instead of eroding and roughening the sea cliff, as Hurricanes Ida and Harvey did, Tropical Storm Bill induced sufficient foreshore sand deposition to cover and protect the cliff. Storm occurrence is clearly not sufficient to infer net erosional processes, and storms can have highly variable effects on local coastal dynamics in this environment, depending

upon sediment supply. Data collected throughout 2015 at Sargent Beach, TX supports the conceptual model that wave-cut platform erosion is controlled by the balance between having (1) enough sand to abrade and erode the platform and (2) too much sand covering and protecting the platform from wave-induced erosion (Sunamura, 1976; Sunamura, 1982; Sklar and Dietrich, 2001, 2004; Walkden and Hall, 2005; Limber and Murray, 2011). Monthly variation in sand cover on the platform is correlated with monthly sea cliff retreat rates during the 2015 survey (Fig. 8), which is evidence for sand cover playing a

critical role in cliff retreat at Sargent Beach.

When large storm events drive considerable roughening of the soft sediment sea cliff, as shown for both Hurricanes Ida and Harvey, subsequent years of smaller storms and fair weather waves then smooth the roughened the sea cliff and with initially high erosion rates. As roughness decreases through time, the cliff approaches a stable morphology—a straight coastline with

an orientation that has characterized Sargent Beach for nearly 150 years (Morton, 1977). Sea-cliff measurements from aerial imagery show a linear relationship between decreasing annual roughness and decreasing annual retreat rates between highly erosive events, such as Hurricane Ida (Fig. 7). After the shoreline returns to its steady-state conditions, cliff retreat and smoothing likely occurs at a relatively slow and steady rate compared to the post-storm condition. We can therefore infer



that the optimal time to implement beach management strategies (i.e., beach nourishment) at Sargent Beach is the recovery
timescale, ~5 years after a storm or shoreline roughening event. However, given the prediction that tropical storms will
increase in intensity the coming years (Emanuel, 2005; Webster et al., 2005), the absence of a roughening storm event on the
Texas Gulf Coast for a 5-year period of time becomes increasingly less likely. Furthermore, we estimate the time to erode
Sargent Beach and breach the Gulf Intracoastal Waterway to be 38 years. To make this estimate, we use only the steady-state
retreat rate calculated from prior data, which does not account for plausible future increases in retreat rate due to roughening
of the shoreline from increased storm activity or sea level rise. Because erosion rates are high at Sargent Beach and there is
little land left between the Gulf of Mexico and the GIWW, this conservative estimate of 38 years represents a serious threat
to coastal communities and the intracoastal waterway.

The two sites studied at Sargent Beach demonstrate how relatively subtle differences in elevation control sea cliff
occurrence. The sea cliff surface is 0.5 m to 1.5 ± 0.2 m higher than the wave-cut platform, which is commonly buried
beneath a sandy beach berm. On the stretches of beach with a lower lying platform, washover fans often develop. On the sea
cliff, sediment instead accumulates at its base, often acting as tools of erosion and filling in hollows and depressions in the
cliff face intermittently before being reincorporated into the shoreface. Additionally, the wave-cut platform underwent an
order of magnitude faster retreat than the sea cliff over the 2015 survey period (Fig. 8). Though elevation changes on this
coastal landscape are small, small changes in elevation cause large changes in position for important plan-view boundaries
due to small coastal slopes. These small elevation changes have implications for the local resilience of the coastline through
the varying erosion rates of the underlying mud substrate and the ability for overwash fans to develop and aggrade narrow
barriers like Sargent Beach, and in general, the coastal plain.

## 6 Conclusions

Storm occurrence and sediment cover jointly control the relationship between cliff roughness and cliff retreat on this
cohesive clay cliff face. Storms that greatly impact the morphology of Sargent Beach are not regular, resulting in long
periods of slow retreat, punctuated by highly erosional events. Between these events, cliff retreat rate first increases with the
initial increase in roughness, then decreases as cliff roughness decreases. Using an empirical model, we calculated a five-
year recovery timescale to the steady-state retreat rate after a roughening event. This may be interrupted by an additional
roughening event, resetting the system before steady-state is reached. Erosion by tropical storms can therefore cause longer-
lasting high erosion rates by roughening the cliff. Changes in monthly cliff face retreat have similar trends to changes in
sediment cover on the wave-cut platform (higher erosion when lower cover and lower erosion when higher cover),
suggesting that the tools and cover effect dominates cliff face retreat at this study site. Observations show that in this
environment, sediment as tools of abrasion may be concentrated on the lateral edges of the headlands, increasing the
sinuosity of the cliff face with high wave action. More work is needed to further quantify the effects of tools and cover in





rocky and soft rock coastal environments. The rapid erosion of this small, soft sediment cliff may be used as a natural laboratory to understand the patterns of erosion on larger cliff systems.

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



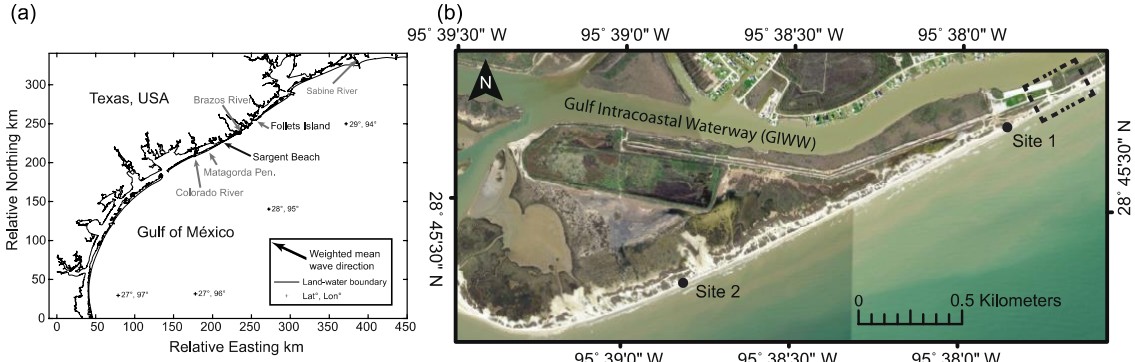

**Figure 1: a) Regional map of the Texas coast showing the location of Sargent Beach. b) Aerial image of two survey sites at Sargent Beach, Texas. Site 1: sea cliff. Site 2: wave-cut platform. The dashed box represents the area where there is the least amount of land between the open ocean and the GIWW.**

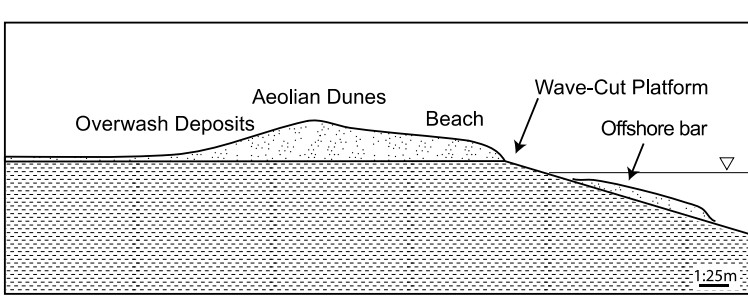

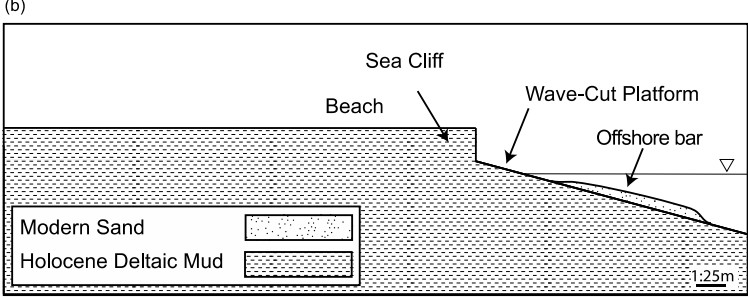

**Figure 2: a) Generalized cross section of wave-cut platform. b) Generalized cross section of wave-cut platform and sea cliff system. ~1:25 m vertical exaggeration.**





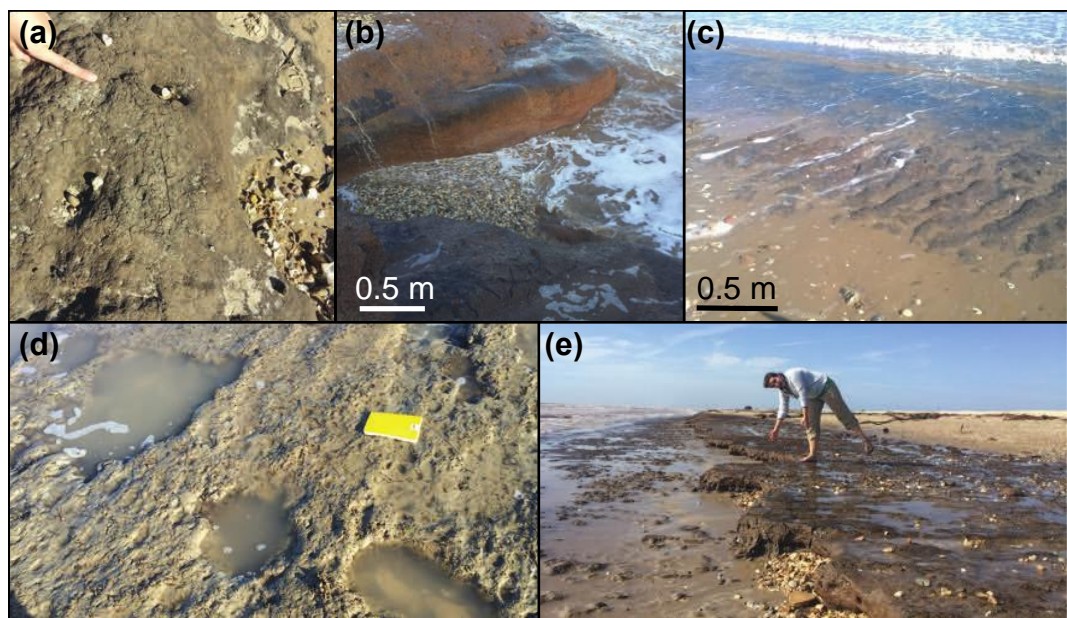

**Figure 3: a) Wetting and drying. b) Focused abrasion creating embayment at the cliff face. c) Focused abrasion creating runnels. d) Potholing and pothole coalescence. e) Differential erosion leading to production of a decimeter scale step.**

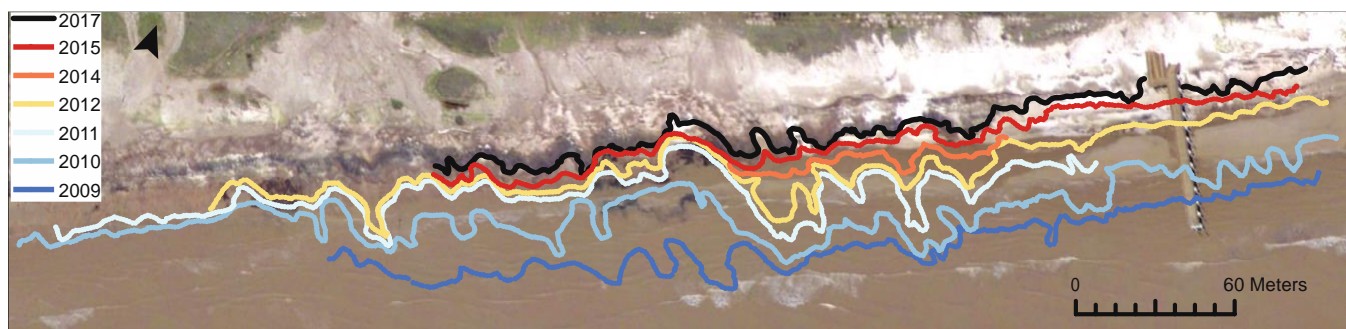

**Figure 4: Plan-view of shorelines delineated from subsequent ~annual aerial images. Site 1 (see Fig. 1). Arrow points north. Image source: NOAA. Hurricane Harvey: Emergency Response Imagery of the Surrounding Regions. 2017.**



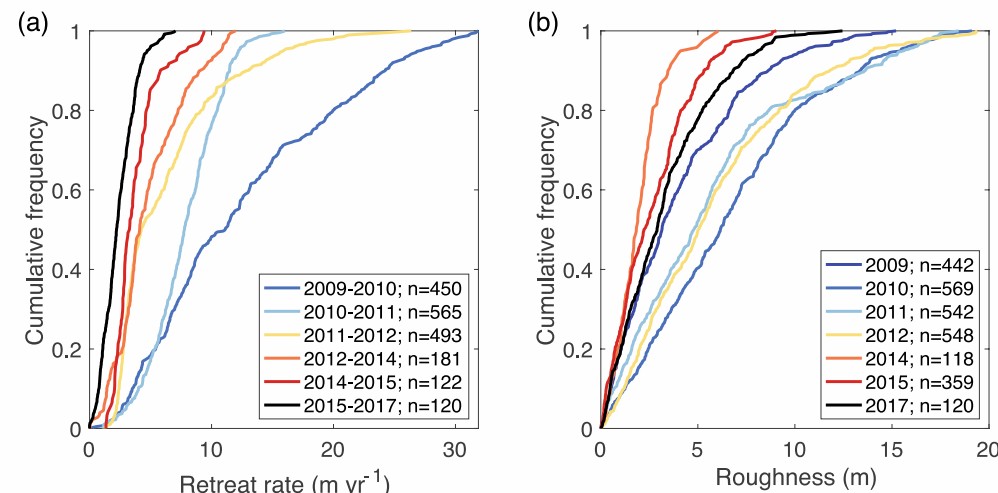

**Figure 5: a) Cumulative distribution function of retreat rate for each time interval. b) Cumulative distribution function of cliff roughness for each time interval.**

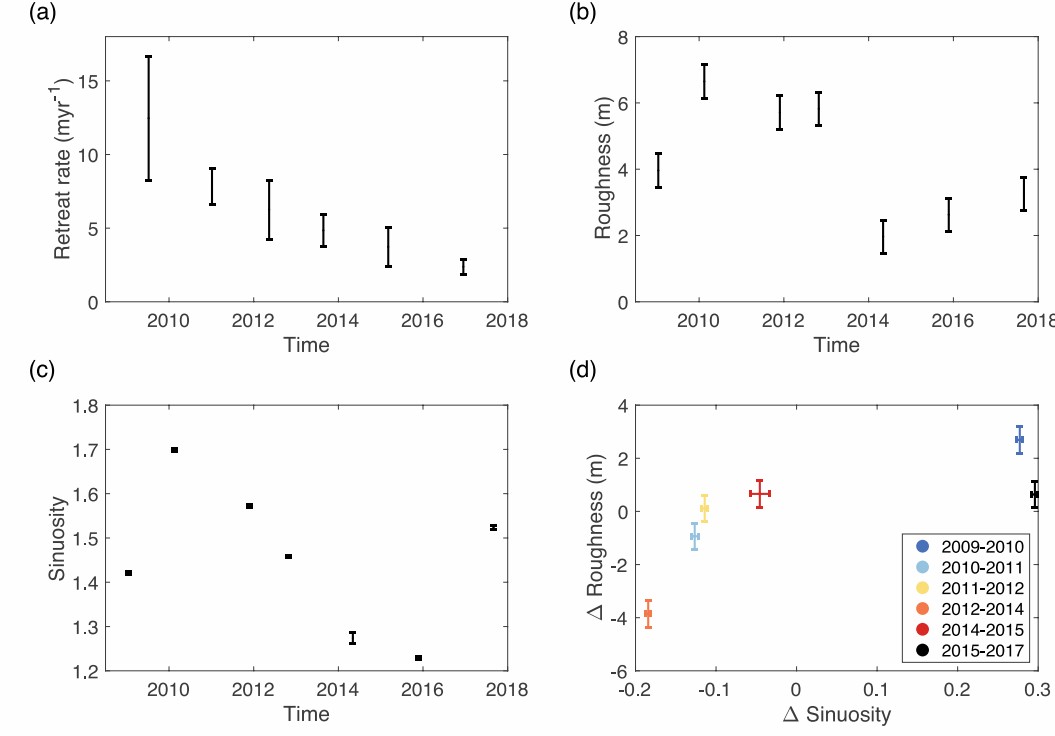

**Figure 6: a) Mean cliff retreat rate (m yr⁻¹) through time. b) Mean cliff roughness (m) through time c) Mean cliff sinuosity through time. d) Change in cliff sinuosity vs change in cliff roughness. Error bars represent propagated error.**



Earth **Surface**
**Dynamics**
Discussions

EGU



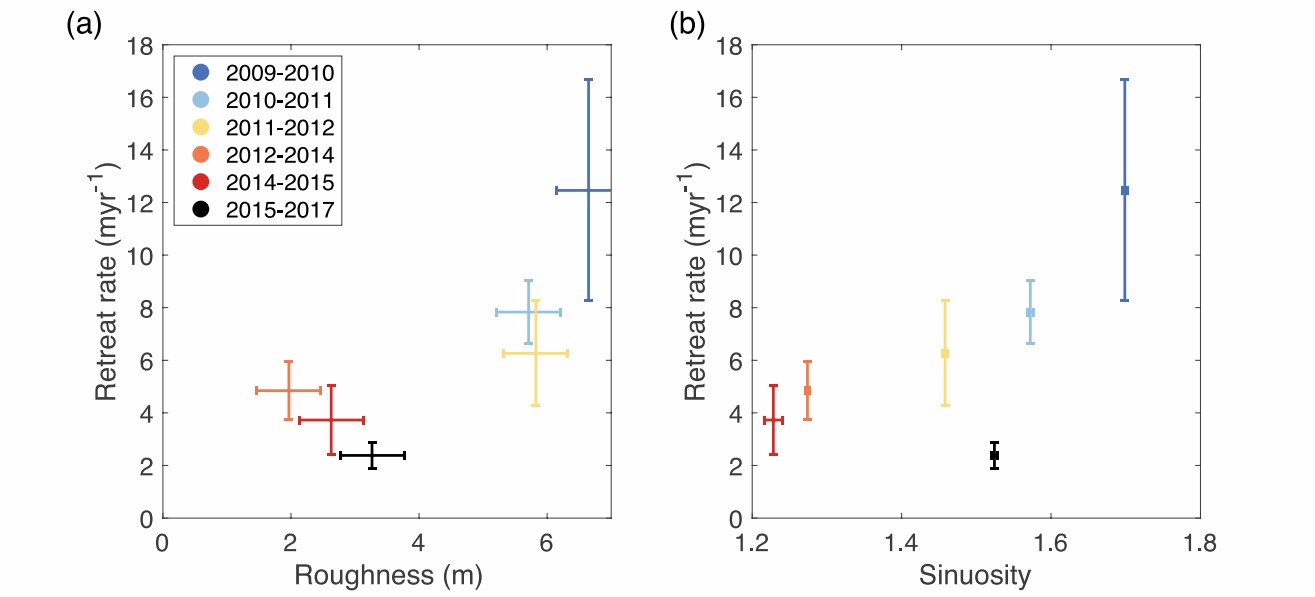


**Figure 7: a) Mean roughness of the second shoreline being differenced vs. mean cliff retreat rate. b) Sinuosity of the second shoreline being differenced vs. mean cliff retreat rate.**

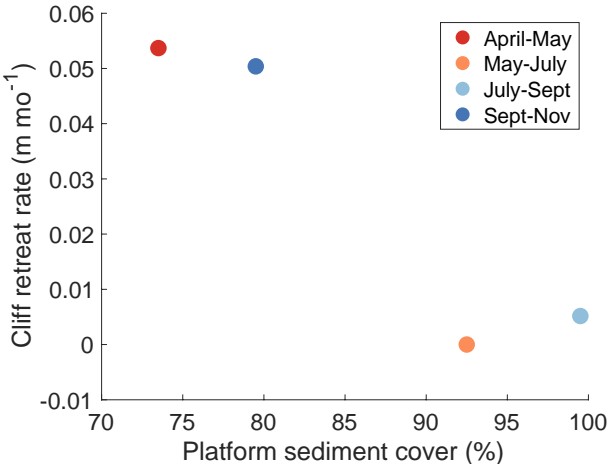

**Figure 8: Platform sediment cover plotted against local cliff face retreat rate in m month$^{-1}$ from monthly surveys conducted for the year 2015.**


Earth **Surface**
**Dynamics**
Discussions

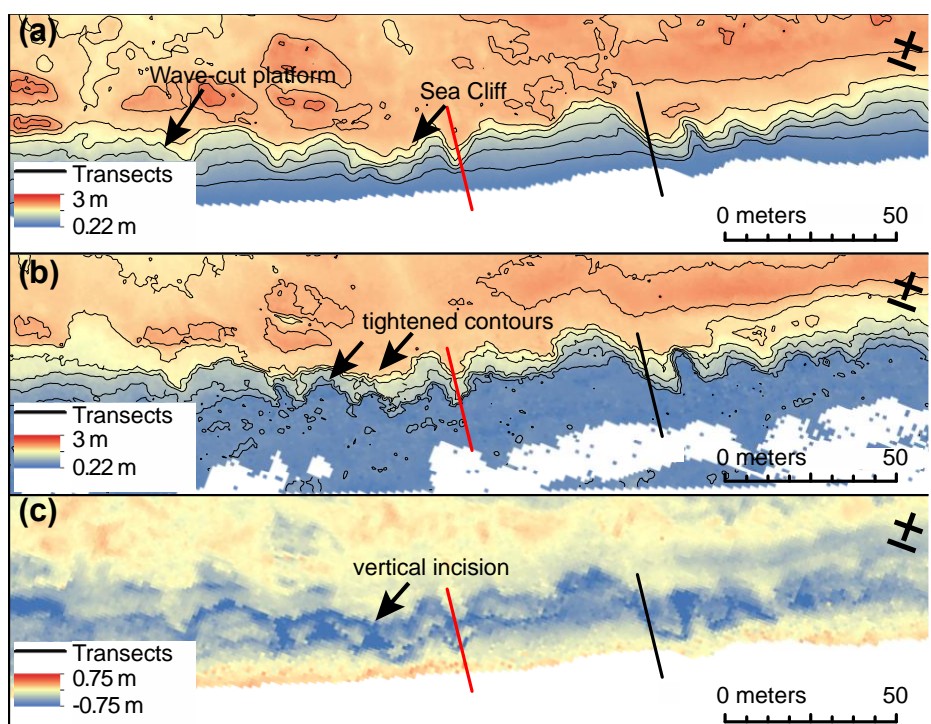

**Figure 9: a) 2016 Lidar collected by the USACE at Site 1. Contour interval is 0.3 m. Transects are indicated by the black solid lines. Arrows point to location on the beach where elevation and contour spacing indicate either the sea**
**cliff or the wave-cut platform. b) 2017 Lidar collected after Hurricane Harvey ©Bureau of Economic Geology. Contour interval is 0.3 m. Transects are indicated by the black solid lines. c) Difference between a) and b). The arrow points to the site of vertical incision.**

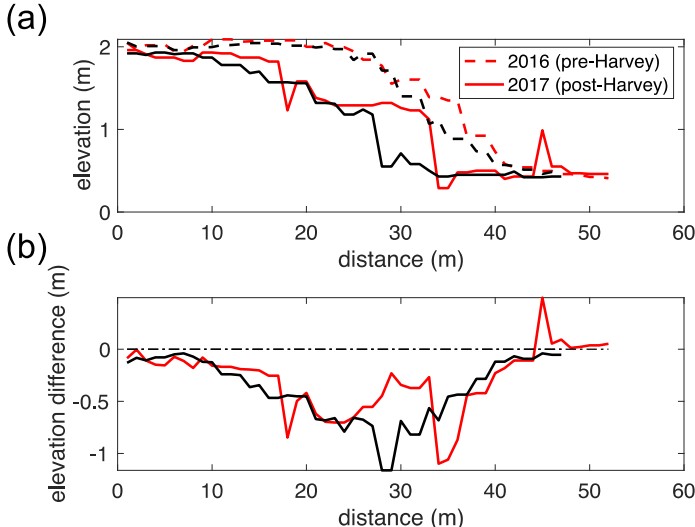

**Figure 10: a) Transects of beach before (dashed lines) and after (solid lines) Hurricane Harvey. b) Difference between 2016 and**
**2017 transects. See Figure 8 for locations of transects at Site 1.**



Earth **Surface**
**Dynamics**
Discussions



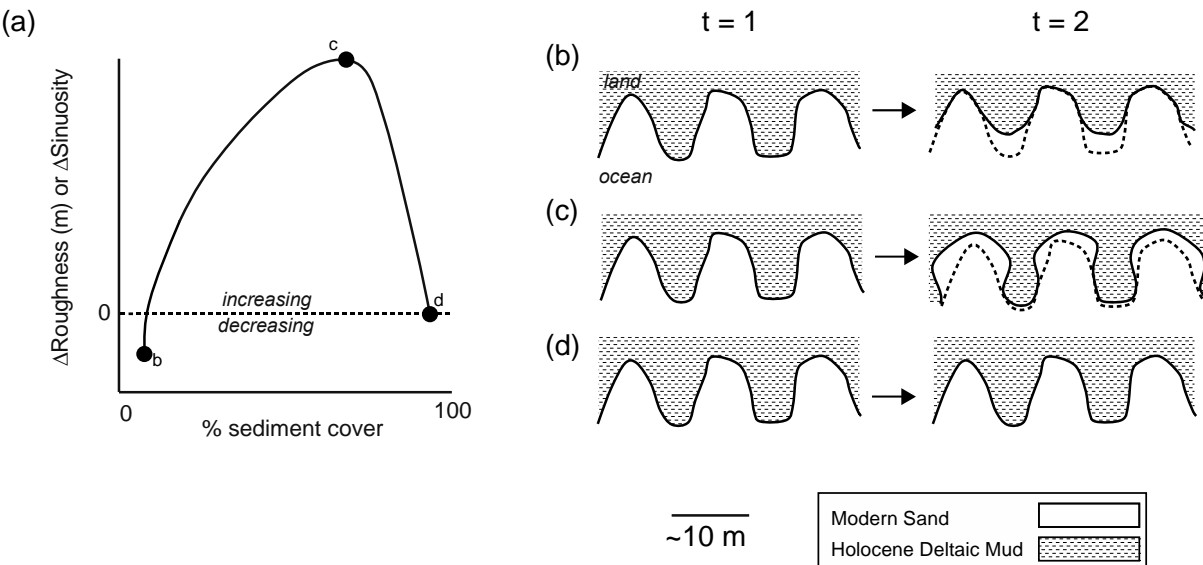

**Figure 11. a) Conceptual diagram of relationship between change in cliff roughness or sinuosity with sediment cover percentage. Positive represents increasing roughness or sinuosity and negative represents decreasing roughness or sinuosity. Planview conceptual diagrams of cliff face erosion when b) no sediment cover, c) intermediate sediment cover, or d) 100% sediment cover. Dashed line represents t = 1 cliff face. Land is at the top and ocean is at the bottom for each panel.**

### Data availability

Data can be found in the supplemental materials and references in the text.

### Author contribution

RP, DM, and AP designed the field data collection and remote sensing analysis. TS conducted the steady state retreat rate modelling. All authors assisted with data analysis and manuscript revisions.

### Competing interests

The authors declare that they have no conflict of interest.

### Acknowledgments

Thank you to Joel Johnson, Sean Gulick, Styze van Heteren, and anonymous reviewers for insightful comments. Thank you to Charlie Kerans, Clark Wilson, and Josh Lambert for assistance with field equipment. And finally thank you to the many field assistants and group members that helped collect and process this data: Alicia Sendrowski, Tim Goudge, Hima Hassenruck-Gudipadi, Ben Cardenas, Wayne Wagner, Kelsi Ustipak, Michael Toomey, and Dylan Rasch. This material is based upon work supported by Plan II at the University of Texas and the National Science Foundation Graduate Research Fellowship under Grant No. 1745302.