# Peer review of "The effects of storms and a transient sandy veneer on the interannual planform evolution a low-relief coastal cliff and shore platform at Sargent Beach, Texas, USA"

_Earth Surface Dynamics, 2021_

## Author Comment (AC1)

Thank you to both of the referees for your thorough and helpful reviews. In our revision we took into account the 2 constructive and thoughtful reviews. All the points raised by the referees were addressed, leading to a more clear motivation and discussion of our dataset. Their initial comments are below in black and our responses are in red.

**Anonymous Referee #1**

This manuscript looks at the occurrence and mechanisms of coastal cliff erosion, using Sargent Beach as a case study. The study is interesting and reveals some good relationships between rates of erosion, storm events, sediment cover, roughness, and sinuosity. My main comments on this manuscript can be addressed by some restructuring and expanding the results. In terms of restructuring, the introduction is lacks theory but some of the background information I was looking for then appeared in the study site section. There is also some study site information that was in the introduction. There is a good dataset behind the results, but I don't feel the authors go into enough detail and then it feels there are big leaps to some of the discussion points and conclusions. I've made some more specific comments below and some minor suggestions/corrections. I hope my comments helps to focus the first half of the paper and draw out some more of the interesting results in the second half.

We appreciate your thorough and helpful comments. We have gone through and addressed these individually, specifically adding detail to the introduction and discussion.

Specific Comments:

1. Intro

- I was looking for a bit more in the intro/lit review before the study site, e.g. move L120-126, 135-142 into the intro – but these need revising so I know why you're telling me all of this. It reads a bit like a list of 'facts' rather than it setting the stage for your research.

Thank you for the suggestion. The introduction was edited to motivate the research more directly. And we restructured the background section to provide more of a review before going into the study site. The subsections are now ordered to first discuss coastal erosion processes and sea cliff evolution (including the lines mentioned above) before Sargent Beach as a setting and the storm history.

- L50 "Although shoreline change at Sargent has been historically analyzed using measurements spaced 50 meters or more apart 50" – can we have some more info on this?

We edited this statement for clarity and added more detail about the cited studies. We also reorganized the paragraph to motivate the study more clearly.

- L54: "Our measurements show", are you talking about results from this study? Shouldn't be in the intro. I'd like more literature/background on this in the intro though .

We reorganized the paragraph to clarify the motivation and moved this to the results section.

2. Methods

- Did you do any estimation of error with your method of detecting the shoreline. It would be good for this to be acknowledge. Was it always really clear where the shoreline/base of the cliff was?

We clarified the text with an explanation that the cliff face was easily identifiable. A description of the estimated uncertainty georeferencing and pixel size can be found on L-164 & L167-168.

- Figure 4 could do with being zoomed out – or having a zoomed out map as well so we can get the context. It's hard to see what exactly we're looking at because of how cropped it is.

Figure 1 was edited to highlight the location of Site 1. Figure 4 will be stacked with an unmapped figure to show the cliff face.

[Figure]

3. Results

- Can you add some values to section 4.1.

Values of erosion rate and sinuosity added to section 4.1. We also added the dates of Hurricanes Ida and Harvey to Figure 6.

[Figure]

- L212 : "Using the fitted empirical model, we calculate a recovery timescale" – please add the results of this calculation – later I read that it's 5 years?

Yes, thank you. Added the value of the recovery timescale here.

- L216: What is the consequence of the erosion pins falling out and only being able to plot the smallest possible retreat? If May-July was actually much larger, this might change the theory that there is little to no retreat for sediment cover >90%

This section was updated for clarity. We estimate a minimum amount of erosion when erosion pins are eroded out of the cliff face entirely. In the alternative case that sediment cover is greater than 90%, the pins were instead buried by deposited sediment. Therefore, we measure no erosion in the case of high sediment cover.

4. Discussion

- You make some quite big statements without, I think, the data to support – or at least we need a little more explanations/acknowledgement of the limitations.

We address the points below to more clearly support our interpretations with the data.

- L246: "Here we show that cliff retreat occurs when sediment cover is <90%" – cliff retreat still ours above 90%, it's just slower. I don't feel you have enough datapoints to identify 90% as a threshold.

The text is updated to be clearer that 90% isn't a specific threshold. Our observations show that high amounts of sediment cover result that future researchers could do a targeted study to quantify a threshold, if it exists, more rigorously.

- L246: "Additionally, our observations of cliff morphology can be linked to erosion by sediment abrasion" – there needs to be more in section 4.1, we don't really see much of the results relating to morphology of the cliff face. I'm not sure where in the results this statement stems from?

The discussion was expanded to support this interpretation using results shown in figures 2b, 4, and 7.

- L247: "If sediment cover is 90% or greater, no cliff erosion occurs and there is no change in morphology" I don't agree the results show this – erosion still occurs >90% even if small and I can't see clearly in the results how this links to morphology? It could become more evident if there was more in results section 4.1.

Edited for clarity to say "If sediment cover is high enough to bury the cliff, little to no cliff erosion occurs and there is no change in cliff face morphology", as was measured when the cliff was buried during the survey period.

- L249: "erosion is focused on the sides and back of the embayments, increasing the sinuosity and roughness of the cliff face" – not sure where this is shown in the results?

We added a clarification of this in the discussion section with a new description of the link between cliff morphology and sediment erosion. See response to L246 comment above. This result is shown in the change in morphology. Our mechanistic explanation for this change in morphology is sediment abrasion.

- L268: "Storm occurrence is clearly not sufficient to infer net erosional processes," – storms can have a significant impact!

Agreed. Edited to say "Although storms can have a significant impact on coastal morphology, storm occurrence alone is not sufficient to infer net erosional processes."

- L284: "at Sargent Beach is the recovery timescale, ~5 years after a storm or shoreline roughening event" – this wasn't shown in the results (see earlier comment).

Added the recovery timescale to section 4.1.1.

Minor comments/Technical Corrections:

L42: please can you define Type-A

Defined as "gently sloping wave cut platform" on L42. Edited for clarity to: "Type-A, gently sloping wave-cut platform (Sunamura, 1992)"

L44: "the largest concentrated extreme of shoreline erosion globally", this sentence doesn't really make sense without reading it a couple of time, although I get the gist. Consider revising this phrasing.

Rephrased to "one of the fastest eroding shorelines globally."

L48: "Similar cohesive coastal cliffs exist globally… London, UK (Hutchinson, 1973)." This is broad and I'm not aware of cliffs in London itself? I could be wrong…

Thank you for catching this error. The London Clay cliffs are found in Walton on the Naze, Essex, not in London. This is corrected.

L59: "and other larger coastal cliff systems" – you only talk about Sargent, so this isn't true. You can only infer?

Correct, we infer. Clarified by changing this to "and potentially other larger coastal cliff systems"

L63: "foreseeable timescale" – do you mean that it is likely to breach more regularly, or that it is likely to breach in the foreseeable future?

In the foreseeable future. Clarified.

L66: "150 m strip of barrier coast that separates the GIWW" …from the Gulf of Mexico.

Added.

L129: "allows sediment particles to act as tools of abrasion" - add 'to'

Corrected.

L261: "Sediment cover is often highest on the cliff face at Sargent Beach in the embayments and  decreases to little or none in front of the headlands." – Can this be revised to be a little clearer – e.g. Sediment cover is often highest in the embayments at Sargent Beach, and lowest at the headlands.

Thank you, revised.

L278: "smooth the roughened the sea cliff and with initially high erosion rate" - remove 'the'

Corrected.

- Gulf Intracoastal Waterway – sometimes you abbreviate this and others you don't.

Changed to abbreviated after the first use.

- Be careful of the length of your sentence – many are 3+ lines long which makes them harder to read and digest.

Thank you for the suggestion. We've gone through and tightened the language throughout.

- Figure 9c: the colours confused me initially as I was expecting red to be negative change. Consider changing this.

Thank you for the comment. Figure 9c and the caption were edited for clarity.

- Abbreviation of TX throughout either needs to be defined or just stated a Texas, for those not in the states.

Changed to Texas throughout.

- Sometimes you use 'five-year' and other time '5-year' – make this consistent.

Corrected to 24 years and changed to consistent formatting.

- Figure 7 in the manuscript text comes before figure 8?
Corrected by switching Figs. 7 and 8

**Juan Felipe Paniagua-Arroyave**

I had the opportunity of reviewing the preprint (esurf-2021-29) entitled "The effects of storms and a transient sandy veneer on the interannual planform evolution a low-relief coastal cliff and wave-cut platform at Sargent Beach, Texas, USA" by Dr. R. Palermo et al., under consideration for publication in the journal Earth Surface Dynamics of Copernicus-EGU. The preprint investigates soft cliff and platform retreat at Sargent Beach (Texas, USA) to understand erosion mechanisms that drive the high retreat at this location. The authors quantify cliff retreat between 2009 and 2017 from 7 aerial photographs using the transects separated alongshore by 1 m. The authors complemented these calculations with a terrain change quantification from lidar-derived digital elevation models of 2016 and 2017. Also, they quantified sediment cover and local cliff erosion (from erosion pins) every 6-8 weeks during 2015. Based on these remote sensing data and field campaigns, the authors quantify

soft-cliff retreat rates, shoreline sinuosity, and shoreline roughness, which they use to drive conclusions and propose a conceptual model of retreat.

The authors focus on: (1) showing that storm events increase shoreline roughness and sinuosity, which in turn drive annual high erosion rates; (2) evaluating the relationship between sediment cover and cliff retreat; and (3) evaluating the temporal and spatial relationship between shoreline roughness and sinuosity with storms and sediment cover. These objectives are interesting from basic and applied standpoints. For example, the authors highlight that understanding cliff retreat at Sargent Beach matters to the local community because it constitutes the barrier that separates the Gulf Intracoastal Waterway from the Gulf of Mexico.

I commend the authors for collecting remote sensing and in situ data to assess shoreline retreat, including relatively novel ways of quantifying cliff erosion (roughness, sinuosity), and providing an insightful discussion on soft cliff retreat. I find the article well written, with sound science and precisely and delicately edited figures. I, therefore, find this article publishable after minor revisions.

I might provide a couple of suggestions worthy of attention from the authors from the stylistic perspective, which I include under "Technical corrections." However, I'd like to highlight a potentially incorrect concept that the authors use profusely in the text: wave-cut platform. As the authors recognize in Line 102 (beginning of section 2.3), subaerial processes may be as crucial as marine (waves) action in driving soft cliff retreat. I suggest replacing the term "wave-cut platform" with "shore platform," which encompasses subaqueous and subaerial drivers. This change shouldn't affect the authors' findings and discussion.

Thank you for the suggestion and thoughtful review. We changed the terminology throughout to shore platform.

Specific comments

   Title: I suggest replacing "wave-cut platform" with "shore platform" here and throughout the text.

Changed. See above.

   Abstract: I suggest including a statement on how did you perform your study (methodology).

Added to the abstract. "using field measurements of sediment cover, erosion, and aerial images to measure shore platform morphology and retreat."

Line 165: I suggest including how did you calculate the uncertainty (e.g., Genz et al., 2007)

The description of the estimated uncertainty due to georeferencing and pixel size can be found on L161-164 & L167-168. We added Genz et al., 2007 as a reference to describe our methodology for calculating retreat rate annually. "For each pair of aerial images, we use the end point retreat rate, measuring the distance between the two cliff face positions and dividing by the time between them (Genz et al., 2007)."

Line 167: Regarding the linear regression, I cannot find Figs. S3-S4. Could you please make sure they are available? Also, why using the linear regression and not using, e.g., the angle related to the principal components or a low pass filter?

Thank you for pointing out this error. These figures were redundant with the text and removed.

We use a linear regression because the overall trend of the shoreline is linear. If this were an overall curving shoreline, this curvature would need to be taken into account and could be done using one of the methods suggested. However, this wouldn't change the results for this study, as a linear approximation is appropriate. An explanation of this was added to the methods section.

Line 170: I suggest showing the equations you used to quantify the sinuosity and roughness.

The definitions of sinuosity and roughness were edited for clarity to: "In the same one meter increments along the exported cliff faces, we calculated local roughness (m) as the distance between the fitted cliff face trendline and the cliff face position. Similarly, pixel size comprises the roughness uncertainty. We also calculate the sinuosity of the cliff face for each image, which is linearly correlated with roughness. The sinuosity is measured as the alongshore distance of the cliff face divided by the distance between the endpoints of the cliff face."

Line 210: I suggest showing the exponential model in Fig. 6a.

The exponential model was added to Fig. 6a.

[Figure]

Line 213: I suggest including the value you obtain for t_r.

The value for t_r was included.

Line 241: this difference is interesting. Any idea why it occurred?

This was removed here and expanded in the discussion. "The long term average of retreat at these locations is similar (Morton, 1977). However, they are approximately two kilometers apart and both have undergone high rates of long term retreat. Although the difference during this field study is large, 5 myr⁻¹ is within the range of previous years' retreat rates of the sea cliff during the study time, with maximum local rates of retreat reaching 25 myr⁻¹. "

Line 244: I suggest changing "coastal" with "cliff" to avoid potential ambiguities related to the statement of sediment being exogenic. This statement could be interpreted as "sediment being exogenic from the littoral cell".

Thank you. Changed.

Line 252: there is a potential feedback occurring (not yet studied) when muddy cliffs erode and the sediment settles during calm wave periods. This settled mud could dampen wave energy, particularly at long (infragravity) periods (Elgar and Raubenheimer, 2008).

Interesting. Thank you for the suggestion. Added.

Technical corrections

- Line 32: I suggest using "cohesive cliffs" instead of "clay cliffs".
  - Replaced clay with cohesive.

- Line 121: please fix the name "Sunamera" in the references.
  - Corrected.

- Line 129: maybe a "to" is missing after "linked"?
  - Corrected.

- Line 136: I suggest checking the writing style of the phrase "Rocky sea cliffs…"
  - Replaced with "Rocky coastal cliffs"
- Line 137: there is a pen-slip in "replated", maybe you mean "related"?
  - Corrected.

- Line 188: please add a space between "1" and "m".
  - Corrected.

- Line 191: I cannot find Fig. S1. Please make sure to include it.
  - We removed the reference to Fig. S1. The figure was removed as unnecessary.

- Line 305: I suggest including a short account of your study in the beginning of the conclusions.
  - Added. "In this study, we measured the retreat rate, roughness, and sinuosity of the cliff face at Sargent Beach over about a decade of aerial imagery. We collected localized measures of cliff retreat, shore platform retreat, and sediment cover in repeat surveys throughout 2015. These data jointly allowed us to explore the relationship between sediment cover, storms, and planform morphology of the cliff face at Sargent Beach. "

- Line 319: I suggest checking References format.
  - References corrected.

- Figure 6: I recommend indicating when Ida and Havey occurred. Also, please separate the units and variables in labels.
  - Ida and Harvey were added to the figures and labels corrected.

- Figure 7: it seems you don't mention this figure in the correct order…
  - Corrected by switching Figs. 7 and 8

- Line 469: you mean black and brown lines?
  - Corrected.

- Line 475: you mean Fig. 9.
  - Corrected.

References suggested

Elgar, S. and Raubenheimer, B., 2008. Wave dissipation by muddy seafloors. Geophysical Research Letters, 35(7).

Genz, A.S., Fletcher, C.H., Dunn, R.A., Frazer, L.N. and Rooney, J.J., 2007. The predictive accuracy of shoreline change rate methods and alongshore beach variation on Maui, Hawaii. Journal of Coastal Research, 23(1 (231)), pp.87-105.

References added.